# Wnt and PI3K/Akt/mTOR Survival Pathways as Therapeutic Targets in Glioblastoma

**DOI:** 10.3390/ijms23031353

**Published:** 2022-01-25

**Authors:** Amir Barzegar Behrooz, Zahra Talaie, Fatemeh Jusheghani, Marek J. Łos, Thomas Klonisch, Saeid Ghavami

**Affiliations:** 1Brain Cancer Department, Asu vanda Gene Industrial Research Company, Tehran 1533666398, Iran; am.barzegar.behrooz@gmail.com (A.B.B.); z.talaie72@gmail.com (Z.T.); 2Department of Biotechnology, Asu vanda Gene Industrial Research Company, Tehran 1533666398, Iran; F.jusheghani@yahoo.com; 3Biotechnology Center, Silesian University of Technology, 44-100 Gliwice, Poland; mjelos@gmail.com; 4Department of Human Anatomy and Cell Science, Rady Faculty of Health Sciences, Max Rady College of Medicine, University of Manitoba, Winnipeg, MB R3E 0V9, Canada; Thomas.Klonisch@umanitoba.ca; 5Department of Pathology, Rady Faculty of Health Sciences, Max Rady College of Medicine, University of Manitoba, Winnipeg, MB R3E 0V9, Canada; 6Department of Surgery, Rady Faculty of Health Sciences, Max Rady College of Medicine, University of Manitoba, Winnipeg, MB R3E 0V9, Canada; 7Department of Medical Microbiology and Infectious Diseases, Rady Faculty of Health Sciences, Max Rady College of Medicine, University of Manitoba, Winnipeg, MB R3E 0V9, Canada; 8Research Institute of Oncology and Hematology, Cancer Care Manitoba, Winnipeg, MB R3E 0V9, Canada; 9Biology of Breathing Theme, Children Hospital Research Institute of Manitoba, University of Manitoba, Winnipeg, MB R3E 0V9, Canada; 10Faculty of Medicine, Katowice School of Technology, 40-555 Katowice, Poland

**Keywords:** glioblastoma, Wnt/β-catenin, PI3K/Akt/mTOR, autophagy, GBM survival

## Abstract

Glioblastoma (GBM) is a devastating type of brain tumor, and current therapeutic treatments, including surgery, chemotherapy, and radiation, are palliative at best. The design of effective and targeted chemotherapeutic strategies for the treatment of GBM require a thorough analysis of specific signaling pathways to identify those serving as drivers of GBM progression and invasion. The Wnt/β-catenin and PI3K/Akt/mTOR (PAM) signaling pathways are key regulators of important biological functions that include cell proliferation, epithelial–mesenchymal transition (EMT), metabolism, and angiogenesis. Targeting specific regulatory components of the Wnt/β-catenin and PAM pathways has the potential to disrupt critical brain tumor cell functions to achieve critical advancements in alternative GBM treatment strategies to enhance the survival rate of GBM patients. In this review, we emphasize the importance of the Wnt/β-catenin and PAM pathways for GBM invasion into brain tissue and explore their potential as therapeutic targets.

## 1. Introduction

Glioblastoma multiforme (GBM) is a type of fatal primary brain tumor in adults, with a median survival duration of about 16 months. Over the previous three decades, survival rates for patients diagnosed with GBM have improved by a mere three months. This is due to the locally aggressive and invasive nature of GBM, the ability of GBM cells to develop therapeutic resistance, and the limited number of drugs capable of reaching therapeutic concentrations in the brain [1,2]. Traditional interventions of surgery, radiation and temozolomide (TMZ) that attempt to reduce tumor mass and attenuate GBM proliferation have failed and have promoted the recurrence of therapy-resistant GBM cells [1,3,4,5]. Because a significant proportion of GBM tumors recur, there is an urgent need for more effective therapeutic strategies [2,6].

Various biochemical changes take place during the epithelium-to-mesenchymal transition (EMT), resulting in the loss of the epithelial structure (at the tissue level) and the development of a non-polar mesenchymal phenotype (at the cellular level). Acquiring a more mesenchymal-like phenotype is associated with greater cellular motility and reduced intercellular adhesion. The latter phenotype promotes migration, invasion, and metastasis [7]. In analogy to other solid tumors, interconnected signaling networks and their downstream genetic regulators have been shown to alter gene transcription and contribute to mesenchymal phenotypes in GBM [6,8]. The wingless/integrated (Wnt)/β-catenin pathway was found to be most active (as assessed by β-catenin nuclear staining) in 30 GBM patient specimens of the mesenchymal subtype [9]. The canonical Wnt pathway is activated when canonical Wnt ligands bind to frizzled (FZD) receptors. This inhibits GSK-3 and prevents β-catenin from being phosphorylated, which results in its ubiquitination and proeasomal degradation. In primary GBM cells, active Wnt/β-catenin signaling increases the expression of EMT activators such as zinc-finger E-box binding homeobox 1 (ZEB1), twist-related protein 1 (TWIST1), and snail-related zinc-finger transcription factor (SLUG1), and promotes the in vitro migration capacity [9,10,11,12]. Importantly, active Wnt/β-catenin signaling has been associated with decreased GBM patient survival. This emphasizes a need for further research to identify the exact roles of canonical Wnt signaling in GBM onset, progression, and recurrence [13]. The phosphoinositol 3-kinase (PI3K)/Akt signaling pathway is instrumental for cell proliferation, stem cell maintenance, and tumor formation [14,15]. Malignant glioma cells produce LIM and SH3 protein 1 (LASP1), LIM, and SH3 protein 1, all of which have been demonstrated to enhance cell proliferation and invasion by activating the PI3K/Akt/snail signaling cascade. The combination of PI3K/Akt inhibitors and TMZ was shown to disrupt the LASP1/PI3K/Akt/snail signaling pathway and significantly suppress tumor progression [16]. In IDH1 mutant glioma, PI3K/mammalian target of rapamycin (mTOR) suppression led to decreased production of the oncometabolite 2-hydroxyglutarate (2HG), and this is associated with improved survival [17].

GBM cells employ the Warburg effect by metabolizing glucose via glycolysis rather than oxidative phosphorylation, even under aerobic conditions, and this is further evidence for the functional involvement of PI3K signaling in GBM cells. The rate of glucose metabolism in human glioblastoma cells is linked to Akt activity, and cells expressing Akt are more vulnerable to mortality after glucose removal. As shown in glioma cells highly expressing Akt, glucose deprivation impairs fatty acid oxidation and causes cell death. Interestingly, stimulation of fatty acid oxidation with 5-aminoimidazole-4-carboxamide ribonucleoside (AICAR) is sufficient to sustain cell viability, indicating that activating Akt impairs the ability of GBM cells to metabolize non-glucose substrates, thus promoting glucose addiction. Akt also contributes to glycolysis regulation by controlling the localization of the glucose transporter 1 (GLUT1) at the plasma membrane and increasing the activity of the key glycolytic enzyme hexokinase (HK). Normally, when growth hormones are removed, cytochrome c is released from the mitochondria, and the caspase cascade is initiated. The pro-apoptotic protein Bax is glucose-sensitive and increasing glycolysis inhibits its activation. Thus, glucose-dependency enables cancer cells with active PI3K/Akt signaling to escape apoptosis in the absence of growth factors [18,19,20,21].

Despite the prevalence and apparent reliance on PI3K signaling in glioma, PI3K inhibitors exert cytostatic rather than cytotoxic effects in glioma cell lines and xenografts even when combined with inhibitors of the epidermal growth factor receptor (EGFR) and mTOR [22]. Thus, the inhibition of PI3K may activate other pathways and processes, such as Wnt/β-catenin and EMT, respectively, both of which enhance cell survival. Combining PI3K inhibitors with drugs blocking these pathways may promote GBM apoptosis.

## 2. Glioblastoma Invasion: Programmed Signaling Pathways

### 2.1. Wnt Pathway

Wnt signaling regulates the self-renewal, proliferation, and differentiation of neural progenitor cells (NPCs) in the brain during different stages of central nervous system (CNS) development. GBM and other cancers (e.g., of the digestive system) have been related to abnormal Wnt pathway activity. Wnt signaling can be subdivided into two categories: canonical and non-canonical (β-catenin independent). A function for the canonical pathway in the maintenance, expansion, and lineage determination of stem/progenitor cell pools has been demonstrated in embryonic and adult tissues. During gastrulation and development, the non-canonical Wnt pathway regulates cell motility and tissue polarity, as well as convergent extension motions of epithelial and neuronal cells [6].

LRP5/6 (low-density lipoprotein receptor-related protein 5, 6) is activated by the canonical Wnt components after binding to cell surface receptors such as FZD. Destabilization occurs when Wnt-binding receptors activate another key complex comprising the APC tumor suppressor gene, glycogen synthase kinase-3 (GSK3), and Axin. This results in the stabilization of β-catenin, which then moves to the nucleus, forms a compound with T cell factor/lymphocyte enhancer factor, and activates Wnt-target genes, including cyclin-D, c-Myc, and VEGF, all of which are essential for embryonic development [23].

The non-canonical pathway can be split into two parts. The disheveled (DVL)-c-Jun N terminal kinase (JNK) pathway/planar cell polarity pathway is involved in cellular polarity. When Wnt binds to FZD via DVL, the JNK kinase pathway is triggered, and this results in cytoskeletal changes. DVL and phospholipase C (PLC) activation by the interaction of Wnt proteins to FZD receptors leads to inositol 1,4,5-triphosphate (IP3) activation through the Ca^2+^-mediated pathway. Activation of protein kinase C, cell division cycle 42 (Cdc42), mitogen-activated protein kinase 7 (MAP3K7), Ca^2+^/calmodulin-dependent protein kinase II (CAMKII), Nemo-like kinase (NLK), and nuclear factor of activated T cell (NFAT) is triggered when IP3 binds to Ca^2+^ channels in the endoplasmic reticulum [23] (see Figure 1).

#### 2.1.1. Wnt Signaling in Glioblastoma

Wnt has a role in preserving the stemness of normal stem cells, enabling repair and regeneration. Dysregulation of Wnt signaling has been reported in several cancers [24]. Mutations in Wnt system components, including APC, β-catenin, AXIN, WTX, and TCF4, may be responsible for Wnt pathway activation in certain malignancies [24]. Colorectal cancers have the highest number of Wnt signaling mutations, and nearly 85% of colorectal cancers have a loss-of-function mutation in APC, while 50% of tumors without an APC mutation have an activating mutation in β-catenin. Mutations were also detected in APC, β-catenin, and AXIN1 in colon cancer and medulloblastoma. In GBM, ovarian, head and neck, breast, and stomach cancers, abnormalities of key components of the Wnt pathway are less common [24]. APC mutations were found in roughly 13% of GBM patients, with a mutation frequency of approximately 14.5%, based on recent studies from a small cohort [19]. Another investigation discovered a homozygous loss of atypical protocadherin FAT1, a tumor suppressor and negative regulator of the Wnt pathway [24]. A dysregulated Wnt pathway may lead to the emergence of cancer stem cells with enhanced aberrant proliferation [25,26,27,28].

#### 2.1.2. Involvement of Wnt Signaling in Glioblastoma Invasion

β-catenin regulates cell–cell adhesion by binding to cadherin as part of the cell adhesion complex. Increased binding of β-catenin to cadherin as well as α-catenin to cadherin or β-catenin promotes glioma cell migration and EMT, and this is mediated by catenin phosphorylation of particular tyrosine residues induced by growth factor signaling. EGFR expression is increased in early GBM, and EGF ligand/EGFR signaling through extracellular signal-regulated kinases 1/2 (ERK1/2) and casein kinase2 (CK2) causes phosphorylation of β-catenin at serine 641 in glioma cells, which has been linked to glioma malignancy [26]. Glioma cell migration and transactivation are aided by β-catenin phosphorylation. Increased expression of the Fz antagonist sFRP2 was shown to prevent glioma invasion by reducing tyrosine phosphorylation of β-catenin and downregulating matrix metalloprotease-2 (MMP-2). Cancer invasion and metastasis have been linked to non-canonical Wnt-5a. For example, Wnt-5a enhances the migration of glioma cells by altering the synthesis of MMP-2 through β-catenin-independent signaling; this includes pathways regulating planar cell polarity and calcium dynamics. U251 astrocytoma cells are less invasive when Wnt-2, Wnt-5a, and Fz2 expression levels are suppressed. Thus, the β-catenin and Wnt signaling pathways seem to regulate selective metalloprotease regulation as an important downstream target to facilitate glioma invasion [26].

#### 2.1.3. Emerging Links between Wnt Signaling and Autophagy in Glioma

An intricate relationship exists between Wnt signaling and autophagy. Canonical Wnt/β-catenin activity inhibits autophagy and the production of the autophagic adaptor p62. Nutrient deprivation promotes the destruction of β-catenin through the autophagic machinery [24,29,30]. Intriguingly, autophagy activation inhibits Wnt/β-catenin signaling in glioblastoma cells through β-catenin subcellular re-localization. The induction of autophagy increases the interaction of β-catenin and N-cadherin in glioma cells and is likely responsible for the formation of newly generated N-cadherin-mediated cell–cell junctions [31]. LC3 targets β-catenin and DVL for autophagic breakdown during nutritional restriction. Interestingly, β-catenin serves as a co-repressor of p62 when Wnt signaling is active. GSK3β is a critical inhibitory regulator of Wnt signaling and has been demonstrated to promote autophagy through phosphorylation of the tuberous sclerosis complex (TSC) [24,32] (see Figure 2).

## 3. Different Forms of Epithelial-Mesenchymal Transition (EMT)

When polarized epithelial cells undergo EMT, they transform into mesenchymal-like cells with a greater capacity to migrate and resist genotoxic substances. Wound healing, embryonic development, and tissue remodeling all require EMT. The EMT process transforms epithelial tumor cells so that they detach from the basement membrane, becoming migratory and invasive without impairing viability. Glioma mesenchymal features may be triggered by similar factors that induce EMT and EMT and play a crucial role in the development of cancer stem cells. Typically, GBM with a mesenchymal subtype behaves aggressively and exhibits markers of neural stem cells. Glioma stem-like cells are extremely invasive and resistant to chemotherapy and radiation in both in vitro and clinical settings [34,35,36]. EMT can be classified into EMT types 1, 2, and 3. Type 1 is necessary for the transformation of primitive embryonic epithelial cells into motile mesenchymal cells, which are necessary for neural crest cell gastrulation and migration. Type 1 EMT plays a critical role throughout embryogenesis in generating morphologically and functionally diverse cell types [37]. EMT type 2 causes the development of excessive fibrosis during wound healing and tissue regeneration [38]. EMT type 3 can be described as a process in which subsets of cancer cells undergo phenotypic alterations that promote invasion, migration, and metastasis. Several studies have indicated that transforming growth factor (TGF) may promote EMT in epithelial cancer cells through the Smad or p38 mitogen-activated protein kinase/Ras homolog family member A pathways [39]. The activation of EMT processes by cues from the tumor microenvironment has also been hypothesized as a key pathway for cancer cells to acquire highly malignant phenotypes. Cancer cells with a transitional mesenchymal phenotype can undergo type 3 EMT and form metastatic tumor nodules at distant locales. While metastasis seldomly occurs in GBM, it is common in EMT [34,40]. In addition, autophagy modulates the transition from an epithelial to a mesenchymal phenotype in non-small cell lung cancer cells through TGFβ 1, and similar processes may be at play in glioma [7,41].

### A Wnt Perspective of EMT in Glioblastoma Invasion

Twist is a transcriptionally active protein involved in cell differentiation and lineage determination. Mesenchymal progression of glioma cells results in the loss of adhesion molecules and cytoskeletal alterations. Twist operates independently of snail to repress E-cadherin and upregulate both N-cadherin and fibronectin through EMT during cancer invasion. Using orthotopic mouse models and brain slice cultures, twist was shown to increase in malignant gliomas and enhance glioma cell invasion via the mesenchymal target gene slug and fibroblast activation protein (FAP). Xenotransplantation experiments revealed that this occurred independently of the cadherin switch. Targeted suppression of twist expression results in a considerable decrease in GBM stem cell sphere formation [42,43,44,45].

E-cadherin expression is regulated by snail, a member of the snail family of transcriptional regulators. Snail also regulates the expression of epithelial (occludins, claudins, and cytokeratin) and mesenchymal markers during EMT (vitronectin and fibronectin). The transcriptional activity of snail is greatly influenced by its subcellular location. After phosphorylation, snail is exported from the nucleus to the cytoplasm and ceases to act as a transcription factor. In response to irradiation, mesenchymal cells release TGF-β, which stimulates snail nuclear localization through the Smad2/3 pathways [34,46,47]. Slug is another transcriptional activator of the snail family, and has been shown to play a key role in inhibiting the epithelial phenotype in a variety of cancer cells. Slug increases the migration and invasion of malignant gliomas. By profiling mRNA expression levels, The Cancer Genome Atlas (TCGA) identified slug and cluster of differentiation (CD)-44 as mesenchymal transition hallmarks in a wide range of cancers [34]. Modulation of autophagy can affect the migratory and invasion capacities of GBM cells by downregulating the EMT proteins snail and slug and, as a result, upregulate cadherin expression [48]. Hyperactivation of the mTORC1/2 complexes hampers autophagy and activates the Wnt pathway, enabling β-catenin to translocate to the nucleus and increase the transcription of pro-invasive factors. Cadherin is expressed and repressed in this situation by EMT players of the SNAI family. Genetic silencing of autophagy-related genes exacerbates the mesenchymal phenotype and increases the capacity of cells to migrate. When autophagy is induced, DVL is destroyed, and the Wnt pathway is inactivated. This results in the cytoplasmic accumulation of β-catenin. SNAI factors are downregulated in autophagic cells and, as a result, N-cadherin accumulates and binds β-catenin [49]. The ZEB proteins ZEB1 and ZEB2 are another notable family of transcription factors that mediate EMT in a variety of tumors, including gliomas [50]. ZEB proteins attach to the E-cadherin promoter region and inhibit the expression of this adhesion molecule, which causes diminished cell–cell interaction and increased tumor cell motility. Recurrences are significantly more common in GBM patients with high ZEB2 levels, and glioma cells become more invasive when activated nuclear factor-B induces ZEB1 expression [34,51,52,53].

RNA-based epigenetic regulation of EMT in GBM is an emerging area with potential therapeutic implications. The long noncoding RNA LINC-PINT inhibits GBM cell proliferation, invasion, and EMT by interfering with Wnt/β-catenin signaling [54]. The miR-101-3p suppresses EMT in GBM by targeting the EMT-promoting factor tripartite motif containing 44 (TRIM44). miR-101-3p suppresses the expression of TRIM44 by directly targeting its 3′UTR [55]. Another microRNA, miR 205, targets ZEB1 via the Akt/mTOR signaling pathway, and miR205 downregulation affects GBM cell motility, invasion, and EMT [56]. Small nucleolar RNA host gene 7 (SNHG7) increases GBM cell proliferation, motility, and invasion by suppressing miR-5095 and by concurrently stimulating the Wnt/β-catenin signaling pathway. miR-5095 and SNHG7 were shown to inversely regulate the Wnt/β-catenin signaling pathway in GBM [57]. Additionally, exosomal miR-133b generated from mesenchymal stem cells inhibits glioma development through the Wnt/β-catenin signaling pathway by targeting EZH2 [58]. Furthermore, miR-128-3p increases TMZ chemosensitivity in GBM by targeting c-Met and EMT [59].

## 4. Therapeutic Approaches to Decrease the Invasion and Survival of GBM Cells by Inhibiting Wnt/β-Catenin and Emt

Adaptive radio-resistance in GBM patients is mediated by the IGF1/N-cadherin/β-catenin/clusterin signaling axis. The accumulation of β-catenin at the cell surface caused by increased N-cadherin expression attenuates the signaling of Wnt/β-catenin and reduces neural cell growth. Transfection of wild type N-cadherin restored radio-resistance in N-cadherin knockout glioblastoma stem cells (GSCs) but not in mutant N-cadherin GSCs missing the β-catenin binding region, which demonstrates the functional significance of the interaction between N-cadherin and β-catenin. Additionally, N-cadherin enhances clusterin secretion to protect GSCs from death after radiotherapy, and N-cadherin deficiency and reduced clusterin release increase the sensitivity of cells to radiation treatment. Radiation was also shown to induce IGF1 secretion, which resulted in enhanced N-cadherin expression, causing an EMT-like phenotypic shift in GSCs [60].

The pharmacological inhibition of the Wnt pathway with the porcupine inhibitor LGK974 acts synergistically with TMZ to decrease glioma cell growth in vitro and is independent of the methylation status of the O6-alkylguanine DNA alkyltransferase (MGMT) promoter [61]. Further transcriptomic analyses revealed that cells treated with LGK974 and TMZ showed substantially decreased expression of aldehyde dehydrogenase 3A1 (ALDH3A1). Suppression of ALDH3A1 expression improved TMZ effectiveness and lowered glioma stem cell clonogenicity, as evidenced by decreased expression of the stem cell markers CD133, Nestin, and Sox2 [61].

Accumulating evidence suggests that TMZ therapy may promote autophagy, which could contribute to GBM therapeutic resistance. The absence of DOC-2/DAB2 interacting protein (DAB2IP) was shown to be responsible for TMZ resistance in GBM through ATG9B. By negatively regulating ATG9B expression, DAB2IP sensitized GBM to TMZ and inhibited TMZ-induced autophagy. ATG9B expression was much higher in GBM than in low-grade glioma. In GBM cells, silencing ATG9B expression inhibited both TMZ-induced autophagy and TMZ resistance. Additionally, DAB2IP decreased ATG9B expression by inhibiting the Wnt/β-catenin pathway. In an attempt to maximize the therapeutic efficacy of TMZ while avoiding therapeutic resistance in GBM, combinatorial therapeutic strategies of TMZ plus a small molecule inhibitor of the Wnt/β-catenin pathway have been proven to act synergistically in GBM cells [62].

Lumefantrine, an FDA-approved antimalarial medication, was recently demonstrated to reverse radiation and TMZ resistance in GBM. Lumefantrine may be a promising GBM treatment option by inhibiting the friend leukemia integration 1 (Fli-1)/HSPB1/EMT/ECM remodeling protein networks. Fli-1 signaling has been implicated in GBM oncogenesis and is overexpressed in radio-resistant and TMZ-resistant GBM, where Fli-1 regulates HSPB1 transcriptionally. Lumefantrine binds Fli-1, and this induces apoptosis in vitro in parental and radio/TMZ-resistant GBM. Moreover, it reduced tumor development in vivo in U87MG glioma cells and radio/TMZ-resistant GBM orthotopic tumor models without systemic side effects [63].

## 5. PI3K/Akt/mTOR (PAM) Signaling Cascade in Glioblastoma Invasion

The extracellular matrix and cytokines transduce their signals via the activation of tyrosine kinase receptors (RTK) and cytokine receptors, which leads to downstream activation of the PAM pathway. PAM is the focus of many cancer studies, since this signaling pathway is frequently overactive in several cancers and has critical roles in cancer cell survival, proliferation, invasion, and migration [64,65,66].

### 5.1. PAM in Gliomagenesis

#### 5.1.1. Role of PAM in Glioma Growth and Invasion

The PAM signaling pathway promotes tumor growth and progression by regulating cell cycle activities. IDH-wildtype GBM has mutations in the growth factor receptor/PI3K/Akt/mTOR pathway [67]. The RTK/PI3K/Akt/mTOR cascade has long been known to boost glioma invasiveness. Upon phosphorylation by Akt, activated mTOR causes cyclin D1 to bind to cyclin dependent kinase (CDK) and initiate cell division. This G1 to S transition can promote carcinogenesis if cyclin D1 is expressed at high levels. Although inhibition of CDK activity by P27kip1 can stall the cell cycle, causing growth arrest, Akt-mediated phosphorylation of P27kip1 neutralizes this critical cell cycle barrier so that cell growth and differentiation are unimpeded. PAM’s powerful metabolic control on the formation of macromolecules, such as proteins, nucleotides and lipids, ensures the rapid growth of cancer cells [66,68,69].

Active PAM signaling networks are operational in the fast majority of GBM [70]. This pathway appears to aid the development of an invasive phenotype of GBM and pediatric brain tumor medulloblastoma (MB) by enhancing motility and stress resistance [71]. The most important PI3K isoform for cell growth and survival is class I_A_ PI3K p110α. In GBM, the gene encoding this isoform, PIK3CA, is often altered [72]. Under anchorage-independent circumstances, the mutant version of PIK3CA plays a key role in cell proliferation in this malignancy. In addition, this PI3K isoform is generally overexpressed in MB where it supports cell proliferation through regulation of the leukemia inhibitory factor receptor α (LIFR α). In MB, inhibiting p110α reduces cancer cell proliferation, migration, and survival [73,74]. Other functionally relevant PI3K class IA isoforms have also been reported in brain tumors. For example, primary GBM overexpress mRNA for p110δ, a class IA isoform that regulates cell motility in these cancer cells [75,76,77]. Other components of the PI3K/Akt pathway are currently being evaluated as possible targets for inhibiting cell growth and migration in GBM and MB. Among these is Akt, which is frequently phosphorylated in malignant brain tumors. This has already led to some promising in vitro observations indicating that Akt inhibition with KP-372-1, KP-372-2, A-443654, or perifosine suppresses cell growth and radio-sensitizes GBM and MB [75,78,79,80].

Phosphatase and tensin homolog (PTEN) inhibits cell growth and interferes with cellular metabolism by downregulating the PAM pathway, whereas inhibiting PTEN activity stimulates Akt and its downstream signaling [81]. In recent work, alterations to the PI3K pathway were found in 44% of the 60,991 solid tumors studied, and PTEN (9.4%) was the second most commonly changed gene after PI3K (13.3%). Changes in PTEN, primarily mutations and profound deletions, are common in uterine, glioblastoma, prostate, lung, and melanoma cancers, according to a pan-cancer restricted analysis of various tumors [82]. Particularly in brain and breast cancer patients, loss of PTEN function has been related to metastasis and a lack of response to radiotherapy and chemotherapy, demonstrating that PTEN is a critical regulator of tumor susceptibility to several treatment options [83,84,85]. Frequently detected in GBM, the level of expression of an inactive form of the tumor suppressor PTEN inversely correlates with tumor aggressiveness. In MB, PTEN is rarely altered, but Akt activity is frequently downregulated [79,86]. PTEN, along with the MAPK signaling cascade, plays a key role in the control of G1/S cell cycle checkpoint-defective astrocytoma invasion, and PTEN loss in GBM cell lines enhances migration, invasion, and resistance to apoptosis [87,88,89]. PTEN regulates Src family kinase activity in a PI3K/Akt-independent manner, and this involves integrin-dependent migration [87,88,89]. Re-expression of functional PTEN in GBM cell lines enhances the cellular amount and activity of the P53 tumor suppressor protein. This causes cell cycle arrest and boosts tumor cell susceptibility to chemotherapeutic drugs, such as the topoisomerase inhibitor etoposide [87,88,89,90].

mTOR has emerged as a key cell growth checkpoint in the PI3K signaling pathway. In the brain, mTOR signaling is mediated by mTORC1 and mTORC2. The mTORC1 signaling route is well recognized due to a wealth of information on mTOR signaling. However, the mTORC2 signaling pathway is less well characterized [91,92,93]. mTORC1 controls cell size and growth in response to nutrition levels, whereas mTORC2 controls cytoskeletal dynamics and the activation of Akt [94,95]. Most GBM exhibit hyper-activation of mTOR signaling [96], and aberrant activation of these particular pathways has been linked to poor prognosis in GBM patients [97]. Tuberous sclerosis complex 1/2 (TSC1/2) is a tumor suppressor. The RAS homolog enriched in brain (RHEB) GTPase activates this complex, which subsequently binds directly to mTORC1 and activates kinase activity [98]. The transcription intermediary factor 1-alpha (TIF-1A) is activated by mTORC1. TIF-1A enables RNA polymerase to transcribe rRNA genes and blocks the MAF1 polymerase III repressor, thus permitting the transcription of 5sRNA and tRNA. mTORC1 also increases protein synthesis and promotes the rapid turnover of lipids and nucleotides typically observed in GBM. Additionally, mTORC1 promotes tumor cell proliferation by suppressing autophagy [99,100]. Unlike mTORC1, mTORC2 cannot sense changes in nutritional state but is activated by growth factors [98]. The activation of various AGC protein kinases by mTORC2 also increases cell proliferation, motility, and survival. In addition to Akt, mTORC2 phosphorylates protein kinase C (PKC)δ, PKCζ, PCKγ, and PKCε, all of which are involved in cytoskeletal construction and cell migration [101,102,103,104,105,106]. Furthermore, the induction of the Warburg effect is mediated by mTORC2 activity. Indeed, mTORC2 stimulates the expression of GLUT4 and the activation of the glycolytic enzymes HK2 and phosphofructokinase-1 (PFK-1) via increasing Akt phosphorylation on serine 473 [107]. Lastly, it was shown that mTORC1 and mTORC2 inhibition may be a promising new treatment for GBM. mTORC2 activity is linked to GBM cell growth and motility. Inhibition of mTOR or PI3K blocks the activation of P70S6K at residue Thr389, which prevents platelet-derived growth factor (PDGF)- or fibronectin (FN)-induced activation of STAT3^Ser727^ upon combined mTOR and PI3K inhibition. Silencing rictor, a co-protein of mTORC2, but not raptor, a co-protein of mTORC1, lowered pAKT expression, and dual mTOR and PI3K inhibition was more effective in reducing cell growth and motility. Pre-treatment with the mTOR inhibitor rapamycin (RAPA) prevented PDGF-induced nuclear localization of mTOR [108].

#### 5.1.2. PAM in Angiogenesis of Brain Tumors

Angiogenesis is a process that involves the formation of new blood vessels and is required for tumors to grow larger than 1 mm in diameter. By providing nutrition and oxygen to tumors, angiogenesis allows tumors to progress and become invasive [109]. GBM tumors have a poor prognosis because they are frequently highly vascularized tumors [110]. Several regulators of angiogenesis promote vascularization during GBM progression, including VEGF, basic fibroblast growth factor (bFGF), hepatocyte growth factor (HGF), platelet-derived growth factor (PDGF), TGF-β, MMPs, and angiopoietins (Angs) [111]. PAM signaling has been identified as a key contributor to GBM angiogenesis [95]. Fibroblast growth factor receptor (FGFR) controls several angiogenic activities, including FGF-induced glioma endothelial cell migration and proliferation, and has been shown to exert its effect on GBM cell survival and angiogenesis through the PAM pathway [112,113]. Like in other cancers, EGFR is active in many GBM and controls tumor formation and angiogenesis [113,114,115]. The RAS/MAPK and PAM signaling pathways have emerged as critical regulators of glioma cell proliferation, differentiation, tumor angiogenesis, and survival in GBM [113,114,115]. It has been demonstrated that rapamycin- and mTOR siRNA-mediated inhibition of the HIF1α and mTOR signaling pathways may decrease the ability of GBM cells to acquire genetic and/or phenotypic characteristics of endothelial cells and form microvascular channels as a way to enhance blood supply to the tumor, called vasculogenic mimicry or trans-differentiation. This supports the notion that mTOR may be a viable therapeutic target in GBM [107]. The triterpenoid celastrol suppresses vasculogenic mimicry formation and angiogenesis in glioma via modulating PAM signaling [116]. The cytotoxic impact of radiation and TMZ in GBM cells is enhanced when targeting the RTK/PI3K/Akt pathway [117]. Loss of PTEN leads to VEGFR2 expression in GBM tumors and may contribute to the failure of anti-angiogenic treatments in GBM [118]. In addition, the overexpression of VEGFR2 in tumor cells might cause early resistance to TMZ and anti-angiogenesis treatment with bevacizumab in GBM patients [119]. In both normal and cancer cells, EGFR stimulates the PI3K/Akt pathway, and EGFR gene amplification in glioma cells results in constitutive PI3K activation [120]. Inhibiting EGFR using EGFR inhibitors PD153035 or AG1478 resulted in the inactivation of this signaling arm [121]. EGFR/PI3K/mTOR inhibitory compounds as well as drugs targeting the Wnt/β-catenin signaling pathways utilized in preclinical and/or clinical investigations for glioma treatment were adapted from [122,123] review papers, respectively (see Figure 3).

## 6. Therapeutic Approaches Targeting PAM in GBM

Several studies have associated over-active mTOR signaling with GBM, and blocking downstream effects mTOR has emerged as a popular treatment strategy [125,126]. The repurposing of several drugs originally intended for other diseases may have promise for treating GBM. The biguanidine drug metformin is typically used to treat type 2 diabetes and acts by activating AMP-activated protein kinase (AMPK) to reduce hepatic glucose synthesis. Activated AMPK is a well-recognized regulator of mTOR activity by phosphorylating and activating TSC2 [127]. There is evidence that metformin has the ability to sensitize GBM and GBM stem cells to TMZ, and this has served as a rationale in an early clinical trial for the treatment of GBM [128,129]. Pimozide is an FDA-approved diphenylbutylpiperidine-class antipsychotic drug that can target STAT5. Pimozide was shown to inhibit Fn14 expression in a STAT5-dependent manner by promoting the TNFR family member FGF inducible 14 (Fn14). Fn14 is known to facilitate cancer cell invasion and survival. One application for pimozide may be as a classical GBM subtype. Excessive STAT5 signaling downstream of constitutively active EGFR variant III (EGFRvIII) was capable of inhibiting the migration and survival of these GBM cells in a STAT5-dependent manner [130]. In addition, the combination of TMZ and pimozide has also been demonstrated to be more effective than TMZ alone.

Chlorpromazine is an antipsychotic medication used for the treatment of schizophrenia and bipolar disorder. Recently, chlorpromazine treatment of C6 glioma cells was shown to arrest the cell cycle in the G2/M phase through transcriptional activation of p21 (Waf1/Cip1) [131]. This transcriptional activation was independent of p53, but involved activation of early growth response-1 (EGR-1). Additionally, chlorpromazine inhibited PAM signaling, resulting in caspase-independent cell death [131]. In other work, TMZ and the combination of endothelial-monocyte-activating polypeptide-ii and miR-590-3p/MACC1 was demonstrated to block PAM signaling in human GBM stem cells [132].

The activation of RTK leads to downstream stimulation of the PI3K/Akt pathway, but RTK inhibitors have shown poor clinical efficacy. It has been proposed that PI3K signaling inhibitors may be helpful in GBM. Buparlisib, a well-tolerated and blood–brain barrier (BBB)-permeable panPI3K inhibitor, attenuates GBM cell growth in vitro and GBM tumor formation in vivo [133,134]. Buparlisib is the most frequently utilized panPI3K inhibitor in clinical studies for GBM therapy, but this drug showed limited efficacy as a single agent in recurrent glioblastoma phase II studies [135]. Sonolisib is an irreversible wortmannin analogue that inhibits PI3K for longer time periods than wortmannin. Sonolisib reduced invasion and angiogenesis in GBM cell lines and improved survival in vivo in orthotopic xenograft models [136,137]. Despite these encouraging preclinical findings, a phase II trial in patients with recurrent GBM had low response rates for sonolisib and no increase in the survival of recurrent GBM patients [138].

The Akt, RAF/ERK, and STAT3 pathways are mainly regulated by EGFR in GBM [139,140]. EGFR mutations increase the invasion and proliferation of GBM cells by activating ERK1/2 and matrix metallopeptidase 1 (MMP1) signaling [139]. Paradoxically, first- and second-generation EGFR inhibitors had limited to no effect on GBM. AZD9291 is the first novel and irreversible third-generation anti-EGFR agent that shows higher efficacy than other EGFR inhibitors in suppressing GBM cell activity and prolonging the life of mice with orthotopic tumor grafts of GBM [141]. While AZD9291 decreases RTK function permanently through covalent EGFR binding, adenosine triphosphate (ATP) analogs and the EGFR inhibitors gefitinib and erlotinib compete reversibly with ATP at the EGFR and require high concentrations to show appreciable drug effects [141,142,143,144,145].

PI3K inhibitors are typically categorized based on their isoform selectivity into pan-PI3K, isoform-selective, and dual PI3K/mTOR inhibitors. At present, approximately 50 PI3K inhibitors are being developed and manufactured for cancer therapy. Table 1 summarizes the characteristics and structural formulae of the most widely used and new PI3K inhibitors. Of note, only a few of these candidates, including BKM120, XL147, and XL765, have thus far entered clinical trials for the treatment of GBM [146,147] (Table 1).

Bevacizumab was approved by the FDA in 2009 for use as a monotherapy or in combination with chemotherapy in the second line treatment of recurrent GBM [176,177]. Bevacizumab inhibits the neovascularization of tumors by neutralizing the impact of VEGF-A but cannot prevent vasculogenic mimicry or the supply of blood from pre-existing vasculature [178,179]. Indeed, histological examination revealed that bevacizumab treatment normalized the vascular structure, reduced microvessel density, and enhanced tumor oxygenation [180]. When used as a first-line therapy, bevacizumab caused an increase in symptom severity, worsening quality of life, and a rapid loss in neurological function [181]. In light of these results, bevacizumab should only be used in individuals who are unresponsive to other treatments. GBM patients likely to relapse during treatment also commonly relapse while treated with bevacizumab [182,183]. GBM recurrence in the absence of VEGF blockade often involves mesenchymal transition in tumor cells, and this is likely mediated by c-Met (a receptor tyrosine kinase of hepatocyte growth factor) activation. In normoxic GBM tissue regions, but not hypoxic regions where VEGF functions are impaired, a functional Met/VEGFR2 complex recovers and accelerates tumor cell invasion and mesenchymal transition [184].

Despite anti-angiogenic therapy increasing tumor oxygenation, hypoxia persists overall in treatment-resistant tumors [180]. The hypoxic microenvironment reactivates HIF-1α, which results in the activation of stromal cell-derived factor 1 (SDF-1) and VEGF and the migration of bone marrow-derived pro-angiogenic monocytic cells, endothelial cells, and pericyte progenitors [185]. Placental growth factor (PGF) is another protein known to promote angiogenesis in the tumor microenvironment by stimulating the VEGF pathways [186,187]. However, when an anti-PGF monoclonal antibody was evaluated in phase I for patients with recurrent GBM, no substantial clinical benefit was observed [188]. The alkylating cytotoxic drug lomustine may have an adjuvant therapeutic effect in recurrent GBM when given in combination with other drugs, as nitrosoureas readily penetrate the BBB [189]. Nonetheless, the impact of chemotherapy is temporary and dependent on treatment dosage and duration until the vasculature returns to normal after anti-angiogenic treatment [190]. An additional treatment option for recurrent GBM is to provide angiotensin system inhibitors (ASI) (given as antihypertensive medications) along with cytotoxic bevacizumab combination therapy [191]. These inhibitors affect both the signaling of the angiotensin II receptor type I (AT1) and angiotensin converting enzyme (ACE). Improved clinical outcomes have been reported in GBM patients receiving standard-of-care (radiotherapy + TMZ) therapy in combination with ASI, but further prospective trials are warranted to validate these observations [192]. Supporting the clinical benefit of ASI and providing mechanistic insight, it was previously demonstrated that inhibiting AT1 with ASI decreased tumor VEGF levels in preclinical glioma models [193].

## 7. GBM Cell Metabolic Functions as a Target of Wnt and PAM Therapeutics

### 7.1. Glucose and Lactate Are Main Fuel Sources for GBM Cells

Non-oxidative glucose metabolism occurs even at normal oxygen levels in cancer cells, while normal cells use the oxidative phosphorylation (OxPhos) pathway to convert glucose into energy [194]. Cancer cells use several different substrates to create raw materials that are needed for cell maintenance and energy production. The importance of these substrates in cancer cell metabolism is becoming increasingly apparent in GBM [195].

Energy metabolization of glucose occurs by means of oxygen-dependent aerobic respiration and oxygen-independent anaerobic fermentation. The most effective form of energy extraction is a chemical process known as OxPhos. The Krebs (tricarboxylic acid, or TCA) cycle is able to use the pyruvate produced during OxPhos as a source of energy for respiring cells. NADH (reduced nicotinamide adenine dinucleotide) fed into the electron transport chain generates ATP at a rate of 36 molecules per glucose molecule. As a result of the Warburg effect, reduction from pyruvate to lactate occurs, and lactate is released into the extracellular space as a byproduct of glycolysis. Only two ATP molecules are produced per glucose molecule, making this procedure a far less efficient way of producing cellular energy equivalents [195]. Even in the presence of sufficient oxygen, cancer cells nearly always adopt this less efficient glycolytic route. It is entirely clear why cancer cells choose to utilize the Warburg effect, but it has been proposed that the amount of ATP is not a limiting factor in cancer cells, considering that this metabolic phenotype manifests prior to the onset of hypoxia [196,197]. Because glycolysis is an inefficient energy-generating metabolic pathway, cancer cells need to consume considerable amounts of glucose and generate large quantities of lactate that decrease the extracellular pH in the tumor microenvironment to acidic (pH ranging from 6.0 to 6.5) conditions. This acidity promotes metastasis, angiogenesis, and, most significantly, immunosuppression, which has been linked to a poor clinical outcome. Thus, lactate might be seen as a critical oncometabolite in metabolic reprogramming during cancer [198]. Cancer cell populations adapt to this process of chronic (several weeks) lactate-induced acidity of the microenvironment by promoting beta-oxidation. This metabolic strategy, known as the Corbet–Feron effect, includes the activation of acetylated proteins and DNA in mitochondria and histone deacetylation in the nucleus. While active in SiHa cervical cancer cells, the FaDu pharynx squamous cell carcinoma line, HCT-116 cells and HT-29 colon cancer cells, GBM cells do not adopt the Corbet–Feron effect. Hence, it is uncertain as to whether this metabolic strategy is functionally relevant in malignant brain tumors [195,199]. However, it has been demonstrated that lactate is a significant alternative energy and biosynthetic substrate that sustains the development of glioma stem cells in the absence of glucose [200].

### 7.2. Role of Fatty Acids in GBM Metabolism

Despite advancements in the genetic characterization of GBM subtypes, metabolic abnormalities that contribute to its aggressive behavior are only now becoming apparent. Some cancer cells, including gliomas, have been demonstrated to have other sources of oxidation in addition to the Warburg effect, suggesting that various substrates are oxidized [196]. Fatty acids may function as important bioenergetic substrates for glioma cells. In particular, primary cultured human glioma cells grown in serum-free medium were shown to oxidize fatty acids to sustain respiratory and proliferative activity. According to results from ^13^C carbon radiolabeling studies in animal models of malignant glioma, more than half of the brain tumor oxidative activity comes from acetate, whereas less than a third comes from glucose [201,202,203]. A recent study suggested that fatty acid β-oxidation (FAO) is the main metabolic node in GBM, providing metabolic flexibility that enables these cells to adapt to their dynamic microenvironment [204]. While glioma cells depend on fatty acids for energy generation, it is currently unclear as to whether these carbon chains derive from the circulation or as a metabolic product from glioma cells themselves. Glucose may be delivered into cells and metabolized into fatty acids via the enzyme fatty acid synthase (FASN); these fatty acids can then be imported into the mitochondria for β-oxidation (futile cycle). In support of this notion, FASN has been demonstrated in glioma cells, and its expression increases with tumor aggressiveness [205].

### 7.3. Amino Acid Metabolism in GBM

Amino acids are a viable nutritional option for the maintenance of glioma. Increased glutathione production by utilizing glutamine may alleviate oxidative and radiation stress and increase chemotherapy tolerance. Glycogen depletion in the liver may be replenished with the help of glutamine, which aids the increased glycolytic rate in cancer cells by increasing blood glucose levels [206,207]. Flushing molecules from the TCA cycle into other metabolic pathways to facilitate tumor growth is possible with a high abundance of glutamine in GBM. Due to a shortage of NADH2/FADH2 for extended OxPhos, glutamine removal may reverse the Warburg effect and lead to cell death in GBM [206,207,208,209]. Deprivation from glycolysis, carbohydrates, or glutamine may be a potential therapeutic option. However, this strategy is not applicable to all GBM. Glutamate generated from glutamine was unexpectedly found to be secreted rather than metabolized. GBM cell development was found to be unaffected by glutamine deprivation because glutamine synthase activity prevented glutamine from entering the TCA cycle. Only a small subset of GBM, mesenchymal in origin and lacking the stem cell marker CD133, might be susceptible to targeting glutamine metabolism. More research is required in this field, especially considering the fact that glutamine is connected to many aspects of cancer metabolism, which may necessitate the deployment of different combinatorial therapeutic strategies for proper metabolic targeting of selected glioma cases [206,207,208,209].

### 7.4. Research Progress on the Changes in Glycolysis and HIF in GBM Invasion

GBM recurrences are frequent and the result of tumor cell infiltration into normal brain tissue which precludes the complete surgical resection of GBM. Bevacizumab is known to enhance the invasion of GBM cells, and bevacizumab-resistant GBM cells are more glycolytic. HIF1 levels were reduced and mitochondrial complex-I levels elevated in invasive versus angiogenic GBM models. Additionally, glycolysis was increased in invasive GBM cells after long-term but not short-term hypoxia. This suggests dynamic metabolic adaptations of GBM cells based on cues from microenvironmental conditions. Emerging evidence suggests an impact of enhanced glycolysis on the invasion and recurrence of GBM [210,211,212]. In vitro, increased glucose-6-phosphatase (G_6_PC) conferred resistance to glycolytic inhibition by 2-DG on GBM cells, while G_6_PC knockdown reduced migration, invasion, and proliferation in vitro and in vivo [213]. Corroborating this result, the expression and activity of glycolytic enzymes were increased in both bulk tumor cells and brain tumor initiating cells from invasive GBM models [214]. A decrease in intracellular ATP results in a rise in the AMP:ATP ratio and activation of the energy-sensing AMPK, which is known to regulate mTORC1 [215]. AMPK functions as an activator of TSC2 and an inhibitor of the mTORC1 component adaptor protein raptor [127,216,217]. In GBM, activation of mTOR signaling results in increased production of transcription factors (such as c-Myc) [218], which in turn increase glycolytic gene expression [219]. Akt also contributes to the glycolytic GBM phenotype by boosting the expression and membrane translocation of GLUT1 and 3 overexpressed in GBM [220,221].

HIF-1 has been implicated in the onset and progression of GBM. Hypoxia increases HIF-1 production, resulting in the transition from aerobic to anaerobic glycolysis, angiogenesis, enhanced cell migratory potential, and genetic changes that prevent hypoxia-induced death [222,223,224]. As a result of the activation of oncogenic signaling pathways, such as PI3K/Akt, MAPK/ERK, and STAT3, HIF-1 transcriptional expression and glucose consumption in GBM are increased, even when oxygen is plentiful [225,226]. HIF-1 activation may promote the production of anti-apoptotic proteins, such as Bcl-2, and thereby inhibit GBM cell death [227,228]. In order to meet the high-energy requirements, HIF-1 acts as a master regulator of aerobic glycolysis in GBM cells while also protecting them from hypoxia-induced damage (see Figure 4 and Figure 5).

## 8. Therapeutic Targeting Wnt and PAM Metabolic Functions to Decrease Invasion and Survival of GBM

Oncogenesis in GBM is thwarted by mTOR inhibitors, which block the activity of mTORC1/2 [230,231]. Metformin and other oral anti-diabetic drugs, as well as chlorpromazine, have beneficial anti-cancer actions by interfering with AMPK/mTORC1 signaling [232]. However, GBM cells can counteract mTOR inhibition by increasing glutamine metabolism via glutaminase (GLS). The combination of mTOR and GLS inhibitors was shown to synergistically reduce GBM tumor formation in a preclinical setting [232]. Hexokinase 2 (HK2), the rate-limiting enzyme that catalyzes the initial step of glycolysis, provides a direct connection between energy metabolism and the autophagy suppressor mTORC1. When glucose shortage occurs, HK2 physically binds to mTORC1 to inhibit its activity and promote protective autophagy [233]. AMPK is required for this process and considered a critical target for reducing autophagy-dependent cancer cell survival in GBM and other gliomas [234,235]. Additionally, the combination of metformin with non-metabolizable 2-deoxy-d-glucose (2-DG) exhibits a powerful inhibitory effect on both glycolysis and oxidative phosphorylation. This resulted in the death of GBM tumor spheres and increased survival in an orthotopic xenograft mouse model [236]. Through acetylation of FOXO and overexpression of c-Myc, mTORC2 regulates glycolytic metabolism in GBM [218]. Metabolic reprogramming of glucose is a significant characteristic of malignant tumors, including GBM, and miRNAs may play a critical role in regulating this process in GBM cells. miR-199a-3p, which specifically targets mTORC2, is significantly downregulated in GBM relative to healthy brain [237,238]. Additionally, miR-34a targets the mTORC2 binding partner rictor in GBM [239]. The tumor suppressor miR-181b may inhibit the growth of glioma cells by blocking SP1-mediated glucose metabolism [240]. Table 2 summarizes repurposed drugs that have shown potential efficacy in the treatment of GBM, along with their hypothesized mechanism of action in tumor cells and associated results.

## 9. Conclusions and Perspective

GBM is known for its invasive nature, frequent recurrence, and poor response to therapy. The inter- and intra-tumoral genetic variants and diverse histological appearance of GBM render this tumor one of the most problematic and challenging cancers. Current treatment of GBM includes surgical excision followed by adjuvant radio-chemotherapy. Despite the improvements in neurosurgical procedures and radiation treatment, the limited efficacy of chemotherapy remains a major challenge. It is important to identify efficient and well-tolerated pharmacological strategies capable of generating lasting responses in specific patient populations, especially with GBM, where frequent recurrence coincide with the emergence of therapeutic resistance. As outlined, the PAM signaling pathways are vital for glioma growth. Preliminary preclinical investigations focusing on the potential effectiveness of PAM inhibitors revealed positive findings. However, thus far, clinical trials have shown little to no benefit from single or combination PAM treatment regimens. Possible explanations for the therapeutic failure of these trials include the limited BBB permeability of particular compounds or the development of GBM resistance or tolerance mechanisms in response to continuous treatment [122].

A significant body of research now supports the claim that Wnt signaling is vital to the pathophysiology of malignant glioma and linked to malignant glioma stem cell proliferation, invasiveness, and treatment resistance. In addition to some promising early clinical findings, there is convincing preclinical evidence of Wnt’s participation in the development of glioma, emphasizing the need for randomized clinical studies on the efficacy of Wnt inhibition in GBM. Advantages of inhibiting Wnt signaling include a reduction in tumor invasiveness, treatment resistance, and tumor cell stemness. Moreover, Wnt inhibition impairs angiogenesis and improves BBB permeability for better delivery of chemotherapy [273,274]. Considering the fact that PAM and Wnt/β-catenin are two of the most important pathways governing cell proliferation and survival in GBMs, targeting PAM and Wnt/β-catenin represents a compelling avenue for developing treatments against gliomas. In this review, we have outlined how the networks composed of PAM, Wnt/β-catenin, and EMT signaling pathways convey information that critically impacts on GBM development. Furthermore, we provided a paradigm for dual-targeted treatments of GBM and suggest potentially promising treatment strategies. More preclinical and clinical studies are required to provide further insights into glioma maintenance, treatment resistance, and recurrence and elucidate how the interactions between PAM, Wnt/β-catenin and EMT affect clinical outcomes.

## Figures and Tables

**Figure 1 ijms-23-01353-f001:**
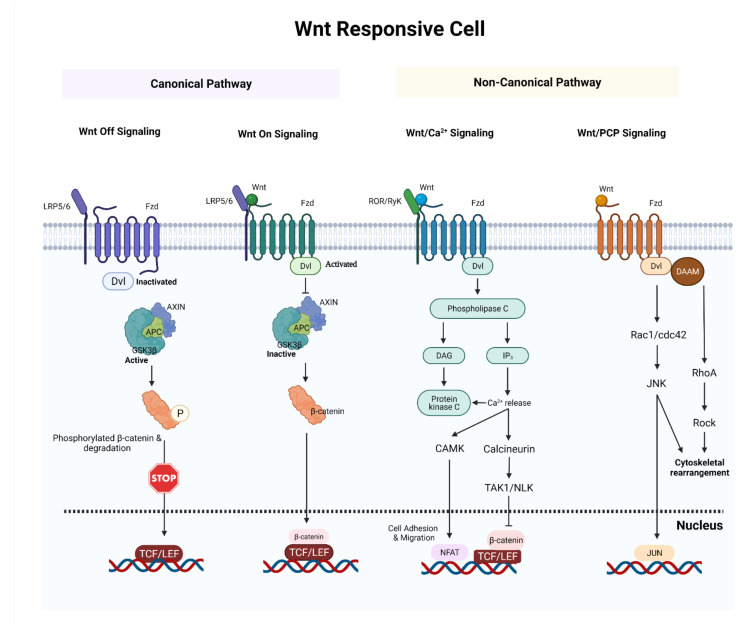
A synopsis of the canonical and noncanonical Wnt signaling pathways. In canonical Wnt off signaling, a combination of AXIN and APC allows GSK3β to phosphorylate β-catenin and target it for proteolysis. In canonical Wnt on signaling, Wnt ligands bind and activate Fzd receptor and activated Fzd receptor recruits Dvl protein and AXIN. This blocks the formation of an AXIN–APC complex and inhibits GSK3β. As a result, β-catenin avoids destruction and accumulates in the nucleus. In non-canonical Wnt/Ca^2+^ signaling, the interaction of Wnt/Fzd with ROR/Ryk co-receptor leads to the formation of IP3 and DAG. DAG activates PKC. IP3 triggers ER Ca^2+^ release which stimulates calcineurin and CAMKⅡ. This activates TAK1/NLK and NFAT. Activated TAK1/NLK can inhibit TCF/β-catenin signaling and NFAT can regulate cell adhesion and migration. In non-canonical Wnt/PCP signaling, the interaction of Wnt/Fzd leads to the recruitment of Dvl, which utilizes its domains (PDZ and DIX) to produce a complex with DAAM. DAAM then stimulates RhoA. Activated RhoA can activate Rock. Dvl can also produce a complex with Rac1 to activate JNK. Abbreviations: APC, adenomatous polyposis coli; GSK3β, glycogen synthase kinase 3β; LRP5/6, low-density lipoprotein receptor-related protein 5/6; TCF/LEF, T-cell factor/lymphoid enhancer-binding factor; IP3, inositol trisphosphate; DAG, diacylglycerol; CAMKⅡ, calcium/calmodulin-dependent protein kinase II; TAK1, transforming growth factor beta-activated kinase 1; NLK, nemo like kinase; NFAT, nuclear factor of activated T-Cells; Dvl; disheveled; Fzd, frizzled; DAAM, disheveled-associated activator of morphogenesis, ROCK, Rho-associated kinase. (Created with https://biorender.com/, accessed on 18 December 2021).

**Figure 2 ijms-23-01353-f002:**
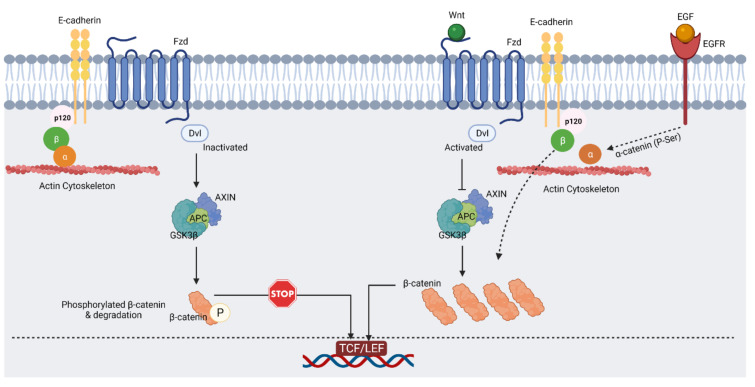
In epithelial cell–cell adhesion and Wnt signaling, the E-cadherin/β-catenin complex plays a critical role. In the absence of Wnt ligand (on the **left**), activated GSK3β can phosphorylate β-catenin, which leads to its proteasomal degradation. As result of that, β-catenin remains in the cadherin/β-catenin/α-catenin complex that is attached to the cytoskeleton (created with https://biorender.com/, accessed on 12 September 2021). In the presence of Wnt ligand (on the **right**), inactivated GSK3β cannot phosphorylate β-catenin, and this results in free β-catenin and its nuclear accumulation. EGF signaling via EGFR, ERK1/2, and CK2 leads to phosphorylation of α-catenin and enhances β-catenin transactivation [26,33]. Abbreviations: EGF, epidermal growth factor; EGFR, epidermal growth factor receptor; ERK1/2, extracellular signal-regulated protein kinase; CK2, casein kinase 2. (Created with https://biorender.com/, accessed on 9 December 2021).

**Figure 3 ijms-23-01353-f003:**
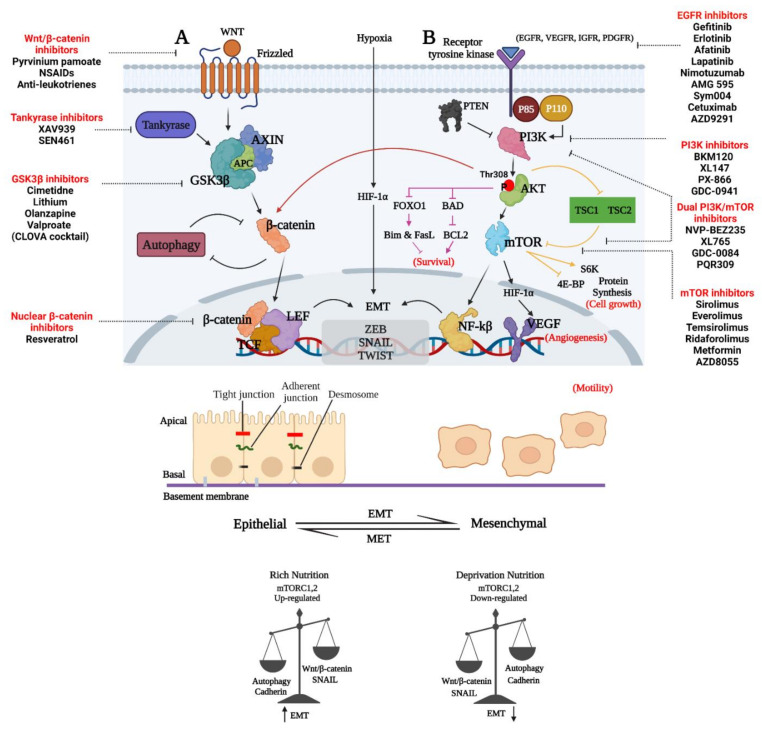
Representative image of GBM invasion through the regulation of the interconnected Wnt/β-catenin, PAM, and EMT signaling pathways. (**A**) Disheveled protein (DVL) enhances the phosphorylation of LRP5/6 co-receptor and the recruitment of axin to the membrane. DVL is recruited in the presence of WNT ligands (WNT1, WINT3A and WNT8) bound to FZD. Non-phosphorylated β-catenin may enter the nucleus when the destruction complex is dissociated to form a complex with the TCF/LEF proteins [124]. (**B**) PI3K regulator (p85) and catalytic isoform (p110) are released from inhibition when growth factors interact with their cognate RTK, resulting in the activation of PI3K. An inhibitor of PTEN may prevent PIP2 synthesis from PIP3, a process catalyzed by PI3K. The PH domain of Akt is then recruited to the plasma membrane by PIP3 and forms an Akt-PIP3 complex. To alleviate the inhibitory effect of TSC1-TSC2 on mTORC1, Akt phosphorylates and inhibits TSC2. Prolonged tumor cell survival, proliferation, migration, invasion, and glucose metabolism are all promoted by activated Akt phosphorylation of downstream system components. The Akt–mTORC1 axis also influences protein synthesis and cell proliferation. An additional negative feedback loop is created when the mTORC1 pathway is activated [122]. Abbreviations: APC, adenomatous polyposis coli; CK1α, casein kinase 1α; GSK3β, glycogen synthase kinase 3β; LRP5/6, low-density lipoprotein receptor-related protein 5/6; β-TREP, beta-transducing repeat-containing protein; TCF/LEF, T-cell factor/lymphoid enhancer-binding factor; TSC1, TSC Complex Subunit 1; 4E-BP, 4E-binding protein 1; FOXO1, forkhead box O1; S6K, ribosomal protein S6 kinase; PDGFR, platelet-derived growth factor receptor; IGFR, insulin-like growth factor 1 receptor; VEGF, vascular endothelial growth factor. (Created with https://biorender.com/, accessed on 25 July 2021).

**Figure 4 ijms-23-01353-f004:**
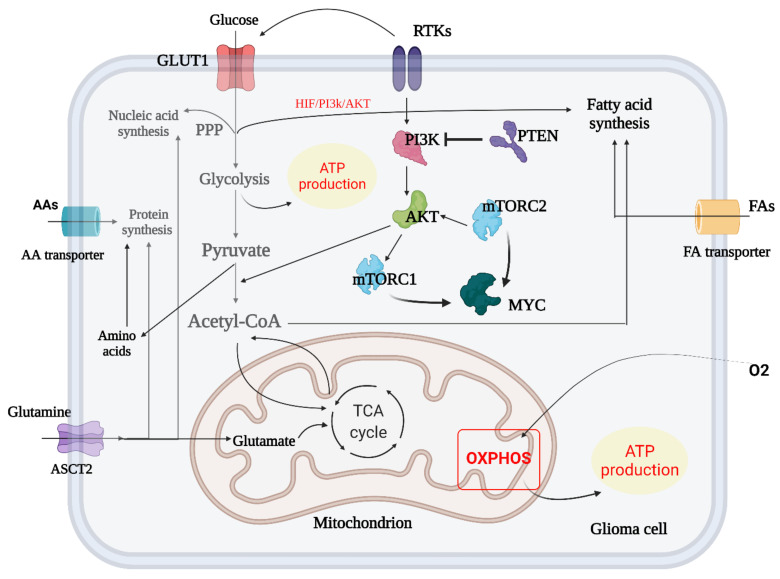
Metabolic reprogramming in glioma. The metabolic switch between glycolysis and TCA cycle in GBM is summarized and amino acid biosynthesis and transport in crosstalk with mTORC1 and mTORC2 are shown. In addition, the crosstalk of fatty acid biosynthesis and TCA/glycolysis cycle is briefly outlined (created with https://biorender.com/, accessed on 14 July 2021).

**Figure 5 ijms-23-01353-f005:**
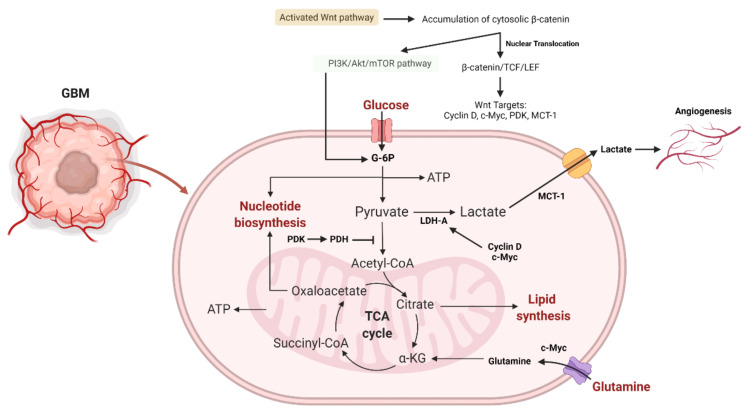
The role of the Wnt and PI3K/Akt/mTOR signaling pathways in glioma aerobic glycolysis. The association of β-catenin to TCF-LEF results in the transcription of Wnt-sensitive genes (PDK, c-Myc, cyclin D, MCT-1). MCT-1 promotes lactate extrusion from the cytosol, hence promoting angiogenesis. Activating PI3K/Akt results in an increase in glucose metabolism. By activating HIF-1α, which inhibits glucose entry into the TCA cycle, Akt-transformed cells defend against ROS damage. HIF-1α-induced PDK1 phosphorylates PDH, culminating in the conversion of cytosolic pyruvate to lactate through the activation of LDH-A. Because PDK inhibits the PDH complex in the mitochondria, pyruvate cannot be converted completely to acetyl-CoA and enter the TCA cycle. Additionally, c-Myc and cyclin D activate LDH-A, which catalyzes the conversion of cytosolic pyruvate to lactate. c-Myc facilitates the entrance of glutamine into the cytosol and mitochondria. Glutamate generated by c-Myc promotes aspartate and nucleotide synthesis [229]. Abbreviations: PDK, pyruvate dehydrogenase kinase; MCT-1, monocarboxylate transporter 1, ROS, reactive oxygen species; PDH, pyruvate dehydrogenase; LDH-A, lactate dehydrogenase A (created with https://biorender.com/, accessed on 28 December 2021).

**Table 1 ijms-23-01353-t001:** PI3K inhibitors and their potential therapeutic use in GBM [146].

Classification	Drug Name	IC50 (nM)						References
		p110α	p110β	p110δ	p110γ	mTORC1/2		
Pan-PI3K inhibitors	Pictilisib	3	33	3	75	580	Inhibits tumor proliferation	[148,149]
	Taselisib	0.29	9.1	0.12	0.97	1200	Inhibits tumor proliferation	[150]
	Buparlisib	52	166	116	H	4600	Apoptotic cell death and p53 deleted cells	[134,151]
	Pilaralisib	39	383	36	23	>15,000	Inhibits tumor proliferation	[152]
	Copanlisib	0.5	3.7	0.7	6.4	45	Induces apoptosis	[153,154]
	Sonolisib	0.1	>300	2.9	N/A	N/A	Complete tumor growth control in the early stages of treatment	[155]
	ZSTK474	16	44	4.6	49	>10,000	Cell apoptosis	[156,157]
	AMG511	4	6	2	1	N/A	Inhibits tumor growth	[158]
Isoform-selective inhibitors	Alpelisib	5	1200	290	250	>9100	Inhibits tumor growth	[159]
	Idelalisib	820	565	2.5	89	>1000	Suppresses GBM cell proliferation and migration	[160,161]
	AMG319	3300	2700	18	850	N/A		[162]
	AZD6482	870	10	80	1090	N/A	PI3K inhibition as anti-platelet therapy; mild generalized anti-platelet effect	[163]
	CH5132799	14	120	500	36	1600	Potent antiproliferative activity	[164]
	AS-605240	60	270	300	8	N/A	Leukocyte activation and migration	[165]
	MLN1117	15	4500	1900	13,390	1670	p110α inhibition; attenuated B cell receptor (BCR)-dependent AKT activation, proliferation, and survival when supported by cytokines BAFF and IL-4	[166]
Dual PI3K/mTOR inhibitors	Dactolisib	4	75	7	5	6	Effective and specific blocking of dysfunctional PI3K activation in human tumor cell models; G1 arrest	[167,168]
	NVP-BGT226	4	63	N/A	38	N/A	p110α inhibition; attenuated B cell receptor (BCR)-dependent AKT activation, proliferation, and survival when supported by cytokines BAFF and IL-4	[166]
	Omipalisib	0.019	0.13	0.024	0.06	0.18/0.3	mTOR inhibitor augmenting anti-proliferative efficacy of PI3K/AKT pathway inhibition	[169]
	Voxtalisib	39	113	41	9	160/910	Anti-proliferative, anti-angiogenic, pro-apoptotic	[170]
	Apitolisib	5	27	7	14	17	mTOR inhibition	[171]
	GDC-0084	2	46	3	10	70	PI3K inhibition	[172]
	VS-5584	16	68	42	25	37	anti-proliferative	[173]
	PF-04691502	1.8	2.1	1.6	1.9	16	Inhibits cell proliferation	[174]
	Gedatolisib	0.4	N/A	N/A	5.4	1.6	Inhibits PI3K/Akt signaling; attenuates proliferation, survival, protein synthesis, and glucose metabolism	[175]

**Table 2 ijms-23-01353-t002:** Licensed drugs targeting Wnt or PAM pathways that have shown potential as repurposed anticancer agents in GBM therapy determined by in vitro and in vivo research and clinical trials [241].

Drug	Suggested Mechanism of Action	IC50	Outcome	References
Chloroquine	Autophagy induction, inhibition of MMP-2 activity, TGF-β secretion and signaling	30 μΜ (U251, LN229, U87MG) 40 μΜ (U251-TMZ^R^, LN229-TMZ^R^, U87MG-TMZ^R^)	Primary cultures of GBM cell lines and specimens showed a 50% reduction in proliferation.	[242,243]
Hydroxy-chloroquine	Autophagy induction	-	Hydroxy-chloroquine eliminated TMZ-resistant glioma cells.	[242]
Mefloquine	Autophagy induction	10 μΜ (U251, LN229, U87) 15 μΜ (U251-TMZ^R^, LN229-TMZ^R^, U87-TMZ^R^)	Mefloquine was capable to killing U251 and U251-TMZ resistant cells at far lower doses than chloroquine.	[242,244]
Quinacrine	Autophagy induction	5 μΜ (U251, LN229, U87) 8 μΜ (U251-TMZ^R^, LN229-TMZ^R^, U87-TMZ^R^)	50 mg/kg of quinacrine substantially slowed the growth of tumors in a subcutaneous human xenograft U87MG model.	[242,245]
Pyrvinium pamoate	Inhibition of Wnt/β-catenin signaling	239.8 nmol/L (BT241), 122.5 nmol/L (BT486)	For 48 h, 200 nmol/L of pyrvinium decreased CD133^POS^ cell fractions in primary (BT428) and recurrent (BT 566) GBM cells.	[246]
Itraconazole	Inhibition of cell proliferation	-	Itraconazole inhibited the proliferation of GBM cells in vitro (2–80 μM, U87MG and rat C6 glioma cells) and in vivo (75 mg/kg, nude mice with U87MG subcutaneous xenografts).	[247]
Salinomycin	OxPhos inhibition in mitochondria	-	Salinomycin decreased the cell viability of GL261 neurospheres and GL261 adherent cells. Salinomycin depleted neurosphere-forming GL261 stem cells from tumorspheres.	[248]
Minocycline	Growth inhibition, autophagy induction, caspase-3 mediated apoptosis	30 μM (C6)	50 μM of minocycline decreased the cell viability of U87MG, U251 and C6 glioma cells; 20 or 100 mg/kg of minocycline (IP) showed slower tumor growth compared controls in Mice injected with C6 cells.	[249]
Chlorpromazine	Inhibition of PAM signaling, autophagy induced cell death	18.8–27.7 μM (C6) 15 μM (SH-SY5Y)	Cell viability was significantly lowered in cells treated with chlorpromazine (≥20 μM) for 24 h. Overall survival greatly improved for U251-TMZ^R^ orthotopic mouse xenograft models.	[250]
Quetiapine	Inhibition of Wnt/β-catenin signaling	-	Relatively high doses of quetiapine (>25 μM) may inhibit cell proliferation by retarding cell cycle in the G2-M phase; 20 mg/kg of quetiapine (IP) alone or combined with TMZ slowed tumor development in orthotopic xenograft mouse model.	[251]
Lithium	Inhibition of GSK-3 activation	-	20 mM lithium for 48 h reduced in viability of 20% of U87MG cells. Through GSK-3 inhibition, lithium concentrations above 5 mM can affect the proliferation, apoptosis and migration of glioma cells. Combination of 1.2 mM lithium and TMZ increased cell death in TP53wt glioma cells and prevented tumor growth in vivo with increased median survival times of mice.	[252,253]
Fluvoxamine	Inhibition of FAK and Akt/mTOR	30 μM	20–30 μM of fluvoxamine inhibited lamellipodia formation, migration and invasion of U87MG and U251 cells in vitro; 50 mg/kg of fluvoxamine inhibited GBM cell invasion and prolonged survival in mice bearing glioma tumors.	[254]
Imipramine	Autophagy induction; inhibition of PI3K/Akt/mTOR	-	60 μM of imipramine was cytotoxic and strongly reduced colony formation of U87MG and C6 cells, but not primary cultured rat astrocytes; 10 μM of imipramine inhibited mitochondrial activity relative to oxygen content in the atmosphere (from 6% in hypoxia, 11% in mild hypoxia, to 19% in medium re-oxygenated at 26% oxygen).	[255,256,257]
Dimethyl fumarate	Autophagy induction	-	-	[258]
Simvastatin	Inhibition of cell growth and migration; inhibition of Ras/ERK and Ras/Akt pathways to induce caspase-3 mediated apoptosis; downregulation of PI3K/Akt	-	10 μM of simvastatin was cytotoxic to U251 and U87MG by inducing aopotosis.	[259,260,261]
Mevastatin, fluvastatin	Inhibition of cell growth; inhibition of Ras/ERK and Ras/Akt; induction of apoptosis	0.922 μM (A 172)	5 and 10 μM of mevastatin and fluvastatin are cytotoxic.	[259,261]
Mibefradil	Inhibition of tumor growth; cell cycle inhibition; activation of pro-apoptotic survivin and BAX pathways; inhibition of Akt/mTOR	-	2.5–5 μmol/L greatly inhibited cell growth and enhanced the inhibition of GSC growth by TMZ; 24 mg/kg (bodyweight) of mibefradil (oral gavage) significantly inhibited growth of tumor.	[262,263]
Losartan	Reducing tumor and cell growth; reduced number of capillary blood vessels; decreased levels of VEGF, PDGF and FGF; apoptosis induction			[193,264]
Metformin	Autophagy and induction of apoptosis; activated AMPK and down-regulation of Akt/mTOR pathway; inhibition of CLIC1 activity with G1 cell arrest	-	10 mM significantly decreased GBM cell proliferation (U87MG, U251, LN18 and SF767).	[265,266]
Pioglitazone	Inhibition of β-catenin expression; reduced cell viability; apoptosis induction	85 μM (U87MG)	100–200 μM significantly reduced the viability of glioma cells (U251, T98G, and U87MG) in a concentration- and time-dependent manner; 100 μM pioglitazone via reducing MMP-2 expression can inhibit U251 cell migration; 50 μM significantly reduced the metabolic activity of G144 cells; 10 μM promoted a minor decrease in metabolic activity in GliNS2 cells.	[267,268]
Aprepitant	Enhanced blockage in Akt phosphorylation due to NK-1 signaling	32 µM (GAMG)	Maximum inhibition at 70 μM after 48 h, with no surviving cells (GAMG glioma cell line).	[269]
Cimetidine	GSK-3β inhibition; reduction in endogenous receptors required for cell adhesion and migration	-	Decreased growth rates of U373 GBM and 9L gliosarcoma cells at concentrations equal to or higher than 100 mM; 100 and 1000 mM cimetidine significantly decreased migration of both cell lines; doses <100 mM did not affect cell cycle kinetics or apoptosis.	[270,271]
Ivermectin	Deactivation of the Akt/mTOR pathways	-	1, 5, and 10 μM inhibited proliferation of U87MG and T98G cells in a dose-dependent manner with ED_50_ of ∼5 μM.	[272]
Rapamycin	Interfering with the AMPK/mTORC1 axis			[108]
Gefitinib and Erlotinib	ATP-competitive and reversible EGFR inhibitors			[141]
Sonolisib	Reduced invasion and angiogenesis in GBM cell lines in vitro; improved survival in orthotopic xenograft models			[136,137]
Buparlisib	Well-tolerated, BBB permeable PI3K inhibitor (most often utilized PI3K inhibitor in clinical studies for GBM therapy)			[135]
Pimozide	Inhibition of migration and survival of GBM cells in a STAT5- dependent manner			[130]

## Data Availability

Not applicable.

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
