# Peer review of "Wnt and PI3K/Akt/mTOR Survival Pathways as Therapeutic Targets in Glioblastoma"

_ijms, 2022, doi:10.3390/ijms23031353_

Round 1
Reviewer 1 Report
Behrooz et al. provide a very nicely written paper highlighting what is known about Wnt signaling and PI3K/Akt/mTOR pathways as a target for glioma treatment. At first I could not understand why the authors picked to review these 2 particular pathways in the same paper, but after reading the paper, I now understand. I do not have any criticisms regarding the paper, it is well written, thorough and easy to follow. Please consider adding the following important papers. (If already referenced, please ignore. You have already referenced so many papers.)
- Cicchini et al. Autophagy regulator BECN1 suppresses mammary tumorigenesis driven by WNT1 activation and following parity. Autophagy, 2014
- Atkinson et al. Activating the WNT/beta-catenin pathway for the treatment of melanoma- application of LY2090314, a novel selective inhibitor of glycogen synthase kinase-3. PLoS One, 2015
- Choo et al. Mind the GAP: Wnt steps onto the mTORC1 train. Cell, 2006.
- Lin et al. Regulation of autophagy of prostate cancer cells by beta-catenin signaling. Cell Phys Biochem, 2015
- Tzeng et al. The pan-PI3K inhibitor GDC-0941 activates canonical WNT signaling to confer resistnace in TNBC cells: resistance reversal with WNT inhibitor. Oncotarget, 2015
- Foltz et al. Epigenetic regulation of Wnt pathway antagonists in human glioblastoma multiforme. Genes Cancer, 2010
- Vadlakonda et al. Role of PI3K-AKT-mTOR and Wnt signaling pathways in transition of G1-S phase of cell cycle in cancer cells. Front Oncol, 2013
- Liu et al. Targeting Wnt-driven cancer through the inhibition of porcupine by LGK974, PNAS, 2013
- Madan et al. Targeting Wnts at the source- new mechanisms, new biomarkers, new drugs. Mol Can Ther, 2015
- Siebzehnrubl et al. ZEB1 pathway links glioblastoma initiation, invasion and chemoresistance. EMBO Mol Med, 2013
- Takebe et al. Targeting cancer stem cells by inhiting Wnt, Notch, and Hedgehog pathways. Nat Rev Clin Oncol, 2010
- Takebe at al. Targeting Notch, Hedgehog, and Wnt pathways in cancer stem cells: a clinical update. Nat Rev Clin Oncol, 2015
- Masui et al. mTORC2 in the center of cancer metabolic reprogramming. Trends Endocrin Metab, 2014
- Chen et al. IBP-mediated suppression of autophagy promotes growth and metastasis of breast cancer cells via activating mTORC2/Akt/FOXO3a signaling pathway. Cell Death Dis, 2013
- Hutt-Cabezas et al. Activation of mTORC1/mTORC2 signaling in pediatric low-grade glioma and pilocytic astrocytoma reveals mTOR as a therapeutic target. Neuro-Oncology, 2013
- Altman et al. Autophagy is a survival mechanism of acute myelogenous leukemia precursors during dual mTORC2/mTORC1 targeting. Clin Can Res, 2014
- Wu et al. Emerging function of mTORC2 as a core regulator in glioblastoma: metabolic reprogramming and drug resistance. Cancer Biol Med, 2014
- Achenbach et al. Synergistic growth inhibition mediated by dual PI3K/mTOR pathway targeting and genetic or direct pharmacological AKT inhibition in human glioblastoma models. J Neurochem, 2020
- Akhavan et al. mTOR signaling in glioblastoma: lessons learned from bench to bedside. Neuro-Oncol, 2010
- Conciatori et al. mTOR cross-talk in cancer and potential for combination therapy. Cancers, 2018
- Daniele et al. Combine inhibition of AKT/mTOR and MDM2 enhanes glioblastoma multiforme cell apoptosis and differentiation of cancer stem cells. Sci Rep, 2015
- Hsieh et al. The translational landscape of mTOR signalling steers cancer initiation and metastasis. Nature, 2012
- Kahn et al. The mTORC1/mTORC2 inhibitor AZD2014 enhances the radiosensitivity of glioblastoma stem-like cells. Neuro-Oncol, 2014
- Masui et al. mTOR complex 2 controls glycolytic metabolism in glioblastoma through FoxO acetylation and upregulation of c-Myc, Cell Metab, 2013
Author Response
Behrooz et al. provide a very nicely written paper highlighting what is known about Wnt signaling and PI3K/Akt/mTOR pathways as a target for glioma treatment. At first, I could not understand why the authors picked to review these 2 particular pathways in the same paper, but after reading the paper, I now understand. I do not have any criticisms regarding the paper, it is well written, thorough and easy to follow. Please consider adding the following important papers.
We thank reviewer 1 for valuable comments
Answer: The mentioned papers were added to manuscript accordingly.
|
Suggested by Reviewer1 |
Revised Manuscript Ref Number |
|
1 |
- |
|
2 |
274 |
|
3 |
256 |
|
4 |
- |
|
5 |
257 |
|
6 |
258 |
|
7 |
259 |
|
8 |
266 |
|
9 |
261 |
|
10 |
262 |
|
11 |
263 |
|
12 |
- |
|
13 |
264 |
|
14 |
265 |
|
15 |
266 |
|
16 |
267 |
|
17 |
268 |
|
18 |
269 |
|
19 |
270 |
|
20 |
271 |
|
21 |
272 |
|
22 |
- |
|
23 |
273 |
|
24 |
200 |
Reviewer 2 Report
In the paper titled "Wnt and PI3K/Akt/mTOR survival pathways as therapeutic targets in glioma," the authors report on Wnt signalling in glioblastoma, therapeutic approaches to reduce invasion and survival, and angiogenesis in brain tumors. The work is systematic and complex with relevant literature.
Minor comments: Section 2 is missing. The numbers should be corrected.
Author Response
We appreciate reviewer 2 for insightful comments.
In the paper titled “Wnt and PI3K/Akt/mTOR survival pathways as therapeutic targets in glioma,” the authors report on Wnt signaling in glioblastoma, therapeutic approaches to reduce invasion and survival, and angiogenesis in brain tumors. The work is systematic and complex with relevant literatures. Section 2 is missing. The numbers should be corrected.
Answer: The numbers were corrected.
Reviewer 3 Report
The authors present an interesting manuscript on the mTOR pathway in glioblastoma. Some aspects need to be considered before accepting the manuscript for publication
-Title: you specifically touch glioblastoma and not glioma in general. Please modify the title accordingly
-Introduction: as per the 2016 update to the WHO CNS classification, "multiforme" was dropped from the definition. please modify accordingly
-metastasis is not really a problem in GBM, while observed in EMT in general. Please clarify to the reader
-After chapter 1 you jump to chapter 3 - please reorganise the manuscript
Apart from these minor aspects, the manuscript is well written and comprehensively addresses all aspects of the field. The figures are in high quality and attention to detail.
Author Response
We respect reviewer 3's critical insights.
Q1: You specifically touch glioblastoma and not glioma in general. Please modify the title accordingly.
A1: The title was corrected accordingly. Page 1, line 2.
Q2: Introduction: As per the 2016 update to WHO CNS classification, multiform was dropped from the definition. Please modify accordingly.
A2: It was corrected accordingly. Page 1, Line 24.
Q3: Metastasis is not really a problem in GBM, while observed in EMT in general. Please clarify for readers.
A3: It was added: While metastasis occurs seldom in GBM, it is common in EMT. Page 6, Line 226.
Q4: After chapter 1 you jump to chapter 3. Please reorganize the manuscript.
A4: The numbers were corrected